# Complete Intradural Interbody Cage Migration in Lumbar Spine Surgery: A Case Report and Literature Review

**DOI:** 10.3390/medicina59050956

**Published:** 2023-05-16

**Authors:** Pang-Hsuan Hsiao, Erh-Ti Lin, Hsien-Te Chen, Yuan-Shun Lo

**Affiliations:** 1Department of Orthopedic Surgery, China Medical University Hospital, Taichung 404327, Taiwan; rayxhiao@gmail.com (P.-H.H.); heyernestlin@gmail.com (E.-T.L.); bonekid1@gmail.com (H.-T.C.); 2Spine Center, China Medical University Hospital, No. 2, Xueshi Rd., North Dist., Taichung 404327, Taiwan; 3Department of Orthopedic Surgery, China Medical University Beigang Hospital, Yunlin 651012, Taiwan; 4Graduate Institute of Precision Engineering, National Chung Hsing University, Taichung 402204, Taiwan

**Keywords:** intradural cage migration, paraplegia, cauda equina syndrome, lumbar spine surgery, case report

## Abstract

*Background*: Spinal fusion is a common surgery, in which vertebrae are fused to restore spinal stability and eliminate pain during movement. The use of an interbody cage facilitates spinal fusion. However, complete cage migration into the dura matter rarely occurs and can be challenging to manage. *Case Presentation*: A 44-year-old man presented at our spine center with a history of incomplete paraplegia and cauda equina syndrome that had lasted for 2 years and 4 months. This condition developed after he underwent six lumbar spine surgeries to address lower back pain and right-sided sciatica. A structural allograft kidney-shaped cage was found completely within the dura at the level of the L3 vertebra. Durotomy, cage retrieval, and pedicle screw fixation from the L2 to L4 vertebrae were performed. Numbness in both lower limbs markedly decreased within several days of the operation. After four months following the progressive physical therapy, the patient could partially control both urination and defecation. Five months postoperatively, he could stand with slight assistance. *Conclusions*: Complete intradural cage migration is a rare and serious complication. To the best of our knowledge, this is the first reported case with such a condition in the literature. Even if treatment is delayed, surgical intervention may salvage the remaining neurologic function and may even lead to partial recovery.

## 1. Background

Spinal fusion is a widely performed surgical procedure, in which the vertebrae are fused to restore spinal stability and eliminate pain during movement. Spinal fusion is typically indicated for degenerative disc diseases, spondylolisthesis, deformities, and fractures that fail to respond to conservative management. In the United States, the number of primary lumbar fusion surgeries performed annually increased by approximately 2.7-fold from 1998 to 2008 [1], and the procedure is becoming increasingly more common worldwide [2]. However, its complication rate is 10–24% [3], substantially higher than that for general orthopedic procedures at approximately 5% [4]. Thus, reducing complications is critical for improving the outcomes of patients undergoing spinal fusion.

The use of an interbody cage facilitates spinal fusion. However, cage-related complications occasionally occur. The complications include misplacement, migration, neurologic injury, and pseudoarthrosis [5]. Among them, interbody cage migration is one of the most common and critical complications, which can result in nerve compression or spinal instability, necessitating revision surgery. Its incidence has been increasing [6]. Herein, we report an extremely rare and challenging case, in which complete cage migration into the dura matter caused incomplete paraplegia and cauda equina syndrome.

## 2. Case Presentation

A 43-year-old male, with a history of six lumbar surgeries for his lower back pain and right-sided sciatica at a local hospital, presented at our spine center with a 2-year-and-4-month history of incomplete paraplegia and cauda equina syndrome. The previous medical history was only partially available. He reported severe numbness in both lower limbs and mostly used a wheelchair for ambulation during the day. Although muscle contraction could be observed in the lower limbs, no functional movements were achieved in the hips, knees, ankles, or toes (MRC grade 1, ASIA grade C). Urinary incontinence and bowel dysfunction were also reported. Rectal examinations revealed poor anal tone. The patient also had type 2 diabetes mellitus and gouty arthritis but no other relevant comorbidities.

Anteroposterior and lateral lumbar spine radiographs revealed pedicle screws in the L2 and L3 vertebrae, one allograft cage in the L3–4 disc, two metallic interbody cages in the L4–5 disc, and, unexpectedly, one foreign body with bone density in the spinal canal at the L3 vertebral level. L2–5 laminectomy and traces of pedicle screws in L4-S1 vertebrae were also observed (Figure 1). Computed tomography scan and magnetic resonance imaging confirmed one interbody cage in the intradural region at the L3 vertebral level, with irregular contours and focal dural thickening (Figure 2). Archnoiditis was also observed at the L2–4 vertebral levels.

After obtaining informed consent, we arranged a salvage surgery for symptom relief. Under general anesthesia, the patient was placed in prone position. Through the previous midline incision in the lower back, we carefully dissected the paraspinal muscles to expose the pedicle screws and rods. The set screws from the L2 to L3 vertebrae and the rods were removed. With the aid of an operating microscope, the fibrotic tissues were removed, and the dura was incised dorsally, and the adhesive tissues between the cage and the dura were gently dissected (Figure 3). The cage was retrieved, and the dura was closed with 6-0 non-absorbable polypropylene sutures (PROLENE, Ethicon, Somerville, NJ, USA). Pedicle screw fixation was then extended to the L4 vertebral level (Wiltrom Spinal Fixation System, Wiltrom, Hsinchu County, Taiwan) to prevent instability. A drainage tube was placed, and the wound was closed.

The patient was transferred to the intensive care unit for one day, and the endotracheal tube was removed the following day. The drainage tube was removed three days after the operation. Rehabilitation programs were arranged, and the patient was discharged after two weeks.

Within several days of the operation, the patient felt a significant decrease in numbness in both lower limbs. He could feel the desire to void in 2 weeks. After four months following the progressive physical therapy, the patient could partially control both urination and defecation. The rehabilitation program mainly included alternative methods between closed-chain and open-chain exercises for lower limbs. Urodynamic studies revealed an average flow rate of 2.8 mL/s and post-voiding residual urine volume of 11 mL (Figure 4). He could also sense bowel movements, and the anal tone was markedly increased, although slightly weaker than normal. Five months postoperatively, he could stand with slight assistance (MRC grade 3^−^, ASIA grade D) (Appendix A), and the rehabilitation program lasted for six months in total. Nerve conduction velocity tests revealed preserved but decreased and prolonged compound muscle action potentials (CMAPs) in the bilateral tibial nerves; no response was observed in the bilateral peroneal nerves. Electromyography revealed positive sharp waves and fibrillation in the bilateral tibialis anterior muscles and gastrocnemius muscles with poor volitional movement. After one year, the patient’s activities of daily living remained unchanged (MRC grade 3^−^, ASIA grade D), and there were no significant complaints of numbness or back pain. Radiographs indicated that there were no signs of implant failures or loosening (Figure 5C,D).

## 3. Discussion and Conclusions

Intradural lesions from outside the dura mater are rare [7]. One case report showed a case with a retropulsed interbody cage at the L3/4 intervertebral area with partial intradural encasement [8]. Though similar to our presented case, the symptoms and the degree of dural involvement differed. The case presented here is remarkable due to the extreme rarity of complete intradural cage migration within the dura matter and the partial regain of the neurologic functions, after more than two years. In 1966, the U.S. Food and Drug Administration approved interbody cages for use in the vertebral disc space; since then, spinal fusion techniques have evolved and now have a high fusion rate [9]. However, cage migration occurs with an incidence of approximately 6.4%, with or without concomitant subsidence [10].

Identifying risk factors related to cage migration or retropulsion is critical. Patient condition, such as osteoporosis [10] and lower body mass index [11], might be associated with cage migration, because they are associated with poor mechanical bone strength and decreased axial compression between the cage and vertebral body. The anatomical structure of the spine also affects the outcomes. Pear-shaped discs are associated with cage migration because of a lack of full contact with all corners of the cage in the sagittal plane [11]. Endplate injury leads to poor endplate preparation and insufficient strength for supporting the interface [10]. Extra stabilization from the pedicle screw fixation is also critical. The standalone interbody fusion cage without pedicle screw fixation shows a higher subsidence rate and lower fusion rate [12].

Moreover, the shape, size, material, positioning, and number of interbody cages can all influence patient outcomes (Table 1). Rectangular or small cages migrate more frequently than kidney-shaped or large cages [6]. Titanium-coated polyetheretherketone (PEEK) cages have superior osseointegration than PEEK-only cages, in which the halo effect was observed at the bone graft–PEEK interface during follow-up [13]. However, the elastic modulus of the cage materials should also be considered; mismatches between the bone and cage elastic moduli could lead to subsidence and further migration [14]. Higher migration rates observed for a posteriorly located cage can also be attributed to the poorer load and strain borne by these cages compared with anteriorly located cages [15].

In this case, no evidence of osteoporosis was observed, and his body mass index was slightly above the normal range (25.4 kg/m^2^). Radiography revealed a common disc type with no apparent convex surface in the posterior halves of the adjacent endplates. The retropulsed cage was a structural allograft (Pinnacle Transplant Technologies, Phoenix, AZ, USA) in the shape of a banana with a threaded design. The cage was 11 mm in height; in our opinion, it was large enough for the L2–3 disc height of the patient, which was measured to be approximately 10.2–11.8 mm in the sagittal plane through computed tomography (not present in the article). During the process of durotomy, the cage was smoothly dissected from the surrounding nerve roots. This was unexpected as the cage was made of an allograft and prone to initiating an immune reaction and, thus, formed the fibrous connective tissue. We postulated that the location of the migrated cage and the abundance of cerebrospinal fluid (CSF) contributed to the situation. After cage retrieval, the patient experienced immediate relief of bilateral sciatica and gradually recovered from cauda equina syndrome. Essentially, cauda equina syndrome is considered to be a surgical emergency, necessitating an emergency surgery within 6 h of its onset [16]; a delay in surgery of more than 48 h is associated with poorer outcomes [17]. However, a case series from India in which decompression surgery was performed with an average delay of 12 days after the onset of cauda equina syndrome demonstrated that some symptom improvement can still be expected [18]. Another case series of patients with posttraumatic syringomyelia revealed clinical improvement after CSF was re-established through subarachnoid–subarachnoid bypass surgery [19]. We believe that, in this case, the direct decompression of the cauda equina within the dura and the consequent restoration of CSF flow contributed to partial neurologic recovery and symptom relief.

In addition to the above academic discussion, the patient still underwent six lumbar spine surgeries with questionable medical decisions and operative techniques. Medical malpractice could lead not only to surgical and anesthetic risks to patients [20] but economic and mental problems also. Careful preoperative evaluation and scanning for the risk factors would help reduce the complications and improve the outcomes. We believe this report to be educational and a reminder that we, as medical practitioners, could always make patients suffer more.

Complete intradural cage migration within the dura matter is a rare but critical complication. Risk factors should be identified for lowering the complication rates. To the best of our knowledge, this is the first reported case with such a condition in the literature. Even if the treatment is delayed, surgical intervention may salvage the remaining neurologic function, and may even lead to partial recovery.

## Figures and Tables

**Figure 1 medicina-59-00956-f001:**
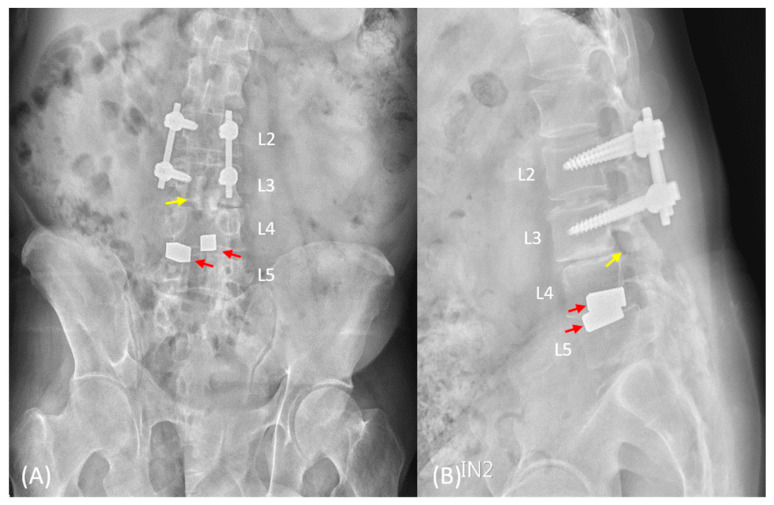
Lumbar spine radiographs taken at our center. (**A**) The anteroposterior view revealed pedicle screw fixation from the L2 to L3 vertebrae, one allograft cage in the L3–4 disc, two metallic interbody cages in the L4–5 disc (red arrow), and, unexpectedly, one foreign body (yellow arrow) with bone density in the spinal canal at the L3 vertebral level. (**B**) The lateral view revealed the shadow of a foreign body cage in the spinal canal at the L3 vertebral level (yellow arrow). Loss of lordosis is also observed.

**Figure 2 medicina-59-00956-f002:**
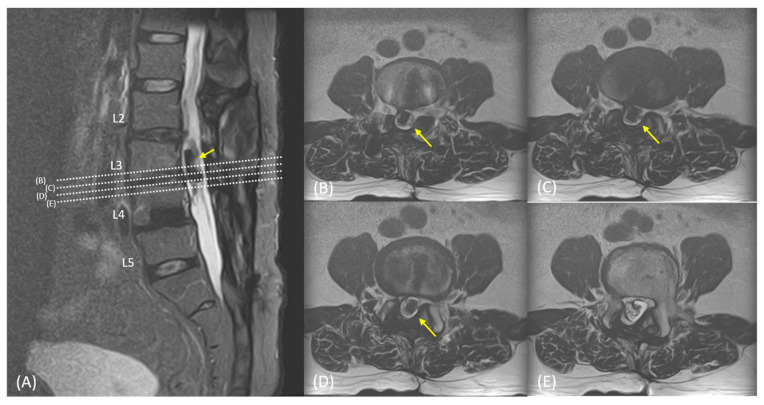
(**A**) Magnetic resonance imaging confirmed one interbody cage (yellow arrow) within the intradural region at the L3 vertebral level, with irregular contours and focal dural thickening. Clumping of the nerve roots was also observed. (**B**–**E**) The axial views corresponded to dotted lines in (**A**).

**Figure 3 medicina-59-00956-f003:**
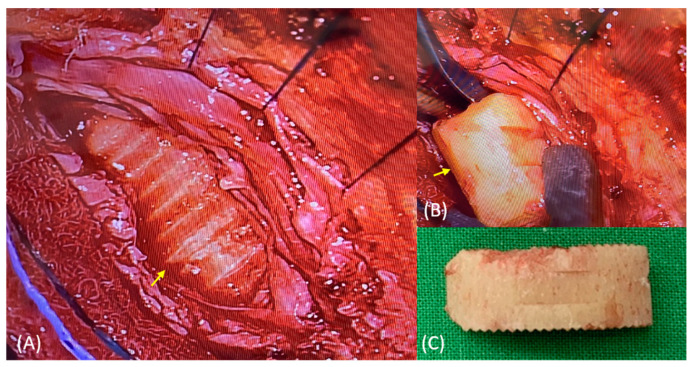
(**A**) Spinal durotomy was performed with the aid of an operating microscope at the L3 vertebral level, and the cage (yellow arrow) was found within the dura. To tack the dural edges, 6-0 Prolene sutures were used. (**B**,**C**) The cage was carefully retrieved with grasping forceps.

**Figure 4 medicina-59-00956-f004:**
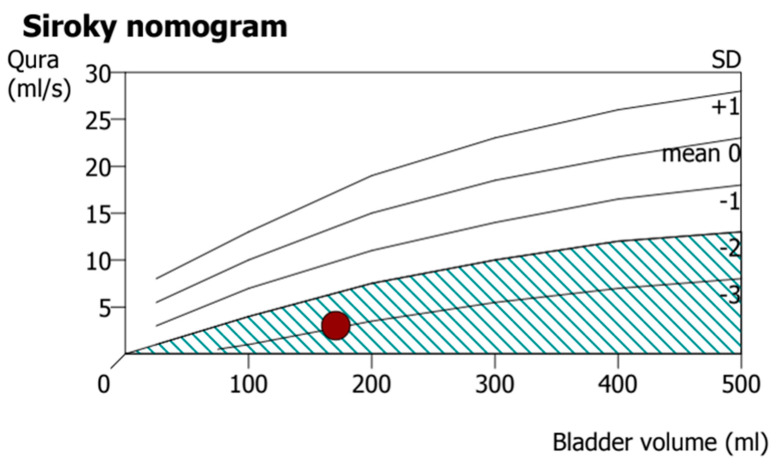
The urodynamic study showed decreased urinary flow rate below two standard deviations.

**Figure 5 medicina-59-00956-f005:**
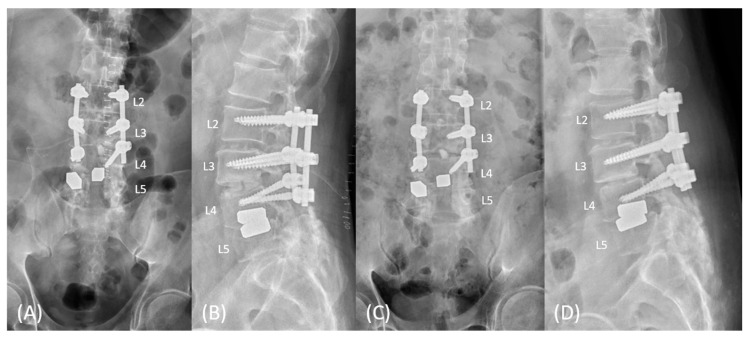
(**A**,**B**) The immediate postoperative anteroposterior and lateral views revealed extended pedicle screw fixation to L4 vertebrae, with the removal of the intradural interbody cage. (**C**,**D**) After a year, the follow-up radiographs revealed no signs of implant failure or loosening.

**Table 1 medicina-59-00956-t001:** Factors influencing the interbody cage migration rates.

**Patient-related**	osteoporosis, BMI
**Anatomy-related**	pear-shaped disc, endplate injury
**Cage-related**	shape, size, material, positioning, the number

BMI, body mass index.

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
