# Peer review of "Complete Intradural Interbody Cage Migration in Lumbar Spine Surgery: A Case Report and Literature Review"

_medicina, 2023, doi:10.3390/medicina59050956_

Round 1
Reviewer 1 Report
Dear Authors,
You are presenting a dramatic case of a patient undergoing 6 surgeries for lumbar pathology in a local hospital without solving the problem but aggravating the situation of the patient.
Please format the manuscript according to MDPI standard.
Insert the author contribution statement at the end of the manuscript.
You present a nomogram in the manuscript, and you also include as supplementary materials, better present it in the manuscript and remove it from the supplementary materials.
Please take into consideration the necessity of discussing the possibility of medical malpractice. Were all surgeries performed under general anesthesia? Were some of the procedures minimally invasive? Please reference this to newer articles from MDPI platform such as the work of Dumitru, M.; Berghi, O.N.; Taciuc, I.-A.; Vrinceanu, D.; Manole, F.; Costache, A. Could Artificial Intelligence Prevent Intraoperative Anaphylaxis? Reference Review and Proof of Concept. Medicina 2022, 58, 1530. Discuss the risks of undergoing so many surgeries under anesthesia.
Moreover, format the reference according to MDPI instructions.
Looking forward to reviewing the improved version of your manuscript.
Dear Authors,
You are presenting a dramatic case of a patient undergoing 6 surgeries for lumbar pathology in a local hospital without solving the problem but aggravating the situation of the patient.
Please format the manuscript according to MDPI standard.
Insert the author contribution statement at the end of the manuscript.
You present a nomogram in the manuscript, and you also include as supplementary materials, better present it in the manuscript and remove it from the supplementary materials.
Please take into consideration the necessity of discussing the possibility of medical malpractice. Were all surgeries performed under general anesthesia? Were some of the procedures minimally invasive? Please reference this to newer articles from MDPI platform such as the work of Dumitru, M.; Berghi, O.N.; Taciuc, I.-A.; Vrinceanu, D.; Manole, F.; Costache, A. Could Artificial Intelligence Prevent Intraoperative Anaphylaxis? Reference Review and Proof of Concept. Medicina 2022, 58, 1530. Discuss the risks of undergoing so many surgical procedures under anesthesia.
Moreover, format the reference according to MDPI instructions.
Looking forward to reviewing the improved version of your manuscript.
Author Response
Question 1:
Insert the author contribution statement at the end of the manuscript.
Response 1:
Dr. Erh-Ti Lin and Dr. Pang-Hsuan Hsiao equally contributed to the design and writing of the manuscript. Prof. Hsien-Te Chen contributed to the investigation and analysis of the findings. Dr. Yuan-Shun Lo revised the manuscript, supervised the work and offered administrative support.
The above was added at the end of the manuscript.
Question 2:
You present a nomogram in the manuscript, and you also include as supplementary materials, better present it in the manuscript and remove it from the supplementary materials.
Response 2:
Thank you for your suggestion. The nomogram is included in the manuscript as ‘Figure 4’ for the integrity of the article. And, the revised figure was attached in the file.
Question 3:
Please take into consideration the necessity of discussing the possibility of medical malpractice. Were all surgeries performed under general anesthesia? Were some of the procedures minimally invasive? Please reference this to newer articles from MDPI platform such as the work of Dumitru, M.; Berghi, O.N.; Taciuc, I.-A.; Vrinceanu, D.; Manole, F.; Costache, A. Could Artificial Intelligence Prevent Intraoperative Anaphylaxis? Reference Review and Proof of Concept. Medicina 2022, 58, 1530. Discuss the risks of undergoing so many surgeries under anesthesia.
Response 3:
Thank you for your suggestion. Indeed, medical ethics is another important issue to discuss with in this particular case. We hope to focus on the surgical and academic portions, and thus enhance the readability of the article. Therefore, we just add a paragraph to be a reminder for the readers as follows in the discussion:
‘Besides the above academic discussion, the patient still underwent six lumbar spine surgeries with questionable medical decisions and operative techniques. Medical malpractice could put not only surgical and anesthetic risks to patients,17 but economic and mental problems. We believed this report be educational and a reminder that we, as medical practitioners, could always make patients suffer more.’
Reference:
- Dumitru M, Berghi ON, Taciuc IA, Vrinceanu D, Manole F, Costache A. Could Artificial Intelligence Prevent Intraoperative Anaphylaxis? Reference Review and Proof of Concept. Medicina (Kaunas). 2022 Oct 26;58(11):1530.
Question 4:
Moreover, format the reference according to MDPI instructions.
Response 4:
Thank you. We’d follow the instructions.
Reviewer 2 Report
Dear Authors,
thank you for submitting your interesting case report "Complete Intradural Interbody Cage Migration in Lumbar Spine surgery: A Case Report and Literature Review". You present a unique and rare case of a 44-year-old man with complete intradural cage migration following multiple lumbar spine surgeries. This is reportedly the first documented case of such a condition, which adds to the existing literature on spinal fusion complications. The case highlights the potential benefits of surgical intervention, even when treatment is delayed, to salvage remaining neurological function and enable partial recovery.
The introduction provides sufficient background on spinal fusion, its indications, and the increasing number of lumbar fusion surgeries. It also highlights the complication rate and the importance of reducing complications. However, the introduction could benefit from more references on interbody cage complications and their prevalence, to better contextualize the presented case.
The case is presented clearly, however following points must be addressed:
· Patient's surgical history: Provide more details about the patient's previous six lumbar surgeries, including the type of surgeries, techniques used, and the timeline of the procedures, to help readers understand the potential contributing factors to the complication.
· Especially describe why there were stand-alone cages used at L4/5. Or was dorsal instrumentation removed?
· Neurological status assessment: The authors should provide a more detailed description of the patient's neurological status pre- and post-revision surgery using a standardized scoring system such as the American Spinal Injury Association (ASIA) Impairment Scale. This would allow for a more objective and comprehensive assessment of the patient's neurological function and recovery, making the findings more informative and comparable with other cases in the literature.
· Add a figure showing, the radiological results after your revision surgery
· Long-term follow-up: Include information on the patient's long-term follow-up to assess the durability of the improvements observed in the case and evaluate the potential need for further interventions.
The discussion paragraph provides a comprehensive overview of the rarity of the case, the factors influencing interbody cage migration rates, and the potential reasons for the patient's recovery. However, there are a few points that can be improved or added:
· More relevant literature: Expand the discussion of similar cases or complications related to interbody cage migration, even if not directly comparable, to provide more context and support for the presented case's uniqueness.
· Post-surgical care and rehabilitation: Discuss the role of post-surgical care and rehabilitation in the patient's recovery process, including specific therapies used and the duration of rehabilitation.
· Recommendations for preventing complications: Provide recommendations for surgical techniques, patient selection, and postoperative management that could help prevent complications like complete intradural cage migration in the future.
· Lessons learned: Discuss the lessons learned from this rare case and their implications for clinical practice and future research in spinal fusion surgery.
The quality of the English language in the provided text is generally good, but there are some areas that could be improved for better clarity and readability:
Sentence structure: Some sentences are quite long and could be broken down into shorter, more concise sentences to improve clarity and readability.
Punctuation: There are a few instances where proper punctuation is missing or misplaced. Carefully proofread the text to correct these errors.
Consistency in terms: Ensure that terms and abbreviations are used consistently throughout the text. For example, consistently use "interbody cage migration" or "cage migration" to maintain clarity.
Word choice: Some word choices could be improved for better clarity and conciseness. For example, replace "the extreme rarity" with "the extremely rare occurrence."
Author Response
Thank you for your opinions and appreciation of this report.
Question 1:
Why are there multiple cages between L3/4 and 4/5 (conventionally there are often 1-2 cages)?
Response 1:
Thank you for your opinion. Before the index surgery, the AP/lateral X-ray revealed one allograft cage in the L3-4 disc, two metallic interbody cages in the L4-5 disc, and unexpectedly, and unexpectedly, one allograft cage in the spinal canal at the L3 vertebral level (shown in figure 1). It’s reasonable that some surgeons apply two PLIF cages in one intervertebral disc space for more stability.
Question 2:
Why were the pedicle screws and rods of L4-S1 removed?
Response 2:
Since we do not have the full documentation and images before the patient came to our hospital, we do not know the rationale of the previous surgeon. However, we speculated PLIF was done from L2 to L5 at least, as the radiographs showed the spinous process and lamina were resected. And, the removal of implant was done, as there was the trace of pedicle screw insertion at L5 and S1 bilaterally. The above was mentioned in the case presentation.
We also speculated the reason of the removal of implant might be due to the numbness in the lower limbs on the both sides persisted, and the previous surgeon might suspect the screws penetrated the pedicles and caused the syndrome. Unfortunately, we do not know the rationale exactly.
Question 3:
The x-ray in Figure 1 does not show the cage deviating into the spinal canal (although it protrudes), whereas the MRI clearly shows the cage in the spinal canal. Has the cage moved due to the change in position? Is the magnetic field of the MRI affecting the cage? Is it because of the interval between imaging? What do you think?
Response 3:
Thank you for your opinion. Figure 1 showed one allograft interbody cage encased in the dural space in the longitudinal arrangement at the area of the L3 spinal canal (indicated as yellow arrow), which was confirmed by the MRI. Please, re-check.
Question 4:
As discussed, inappropriate cage size selection may be a contributing factor, but I also thought that the cage may have migrated due to stress concentration between the L3/4 vertebrae as a result of the removal of posterior fixation at L3-S1. (Often, even though the intervertebral bodies appear to be fused, they are in fact not fully cross-linked.) I thought it would be worth mentioning that at the very least, inadvertent implant removal can lead to subsequent implant-related complications.
Response 4:
Thank you for your opinion. Besides the cage material, shape, size, and the number, …, the extra stabilization is also critical for the vertebrae to be fused, especially during the early stage.
We supplemented in the discussion as follows:
Extra stabilization from the pedicle screw fixation is also critical. The standalone interbody fusion cage without pedicle screw fixation shows higher subsidence rate and lower fusion rate.11
- Zhang J, Pan A, Zhou L, Yu J, Zhang X. Comparison of unilateral pedicle screw fixation and interbody fusion with PEEK cage vs. standalone expandable fusion cage for the treatment of unilateral lumbar disc herniation. Arch Med Sci. 2018 Oct;14(6):1432-1438.
Reviewer 3 Report
The authors have reported on a rare & interesting case of intradural migration of interbody cage . The case report is of interest & can be considered fro publication .
general comments
1) needs overall English language editing
2)patient consent
3) minimum duration of followup - ideally is 1 year . Along with a follow up X-ray & outcome details
specific comments
1) neurological examination should be in MRC grade - both at presentation & follow up
2) why was CT scan not performed .
needs English language. editing
Author Response
Thank you for your suggestions and appreciation of this report.
Question 1:
Needs overall English language editing.
Response 1:
Thank you for your suggestion. We’ve reviewed and revised the manuscript accordingly. Please, re-check and reconsider the quality of English in the article.
Question 2:
Patient consent.
Response 2:
The informed consent was already sent to the assistant editor. Please, re-check.
Question 3:
Minimum duration of followup - ideally is 1 year. Along with a follow up X-ray & outcome details.
Response 3:
Thank you for your suggestion. We complement the followup in the case presentation as follows:
After a year, the activities of daily living remained still (MRC grade 3-), and no markedly disturbing numbness or back pain was reported. The radiographs showed no signs of implant failure or loosening (Figure 5C and 5D).
Figure.5
(A) and (B) The immediate postoperative anteroposterior and lateral views revealed extended pedicle screw fixation to L4 vertebrae, with removal of the intradural interbody cage. (C) and (D) After a year, the followup radiographs revealed no signs of implant failure or loosening.
Question 4:
Neurological examination should be in MRC grade - both at presentation & follow up.
Response 4:
Thank you for your suggestion. Before the index surgery, the physical examination revealed MRC grade 1 in the lower limbs on the both sides. Five months after the surgery, his muscle strength improved to MRC grade 3- and persist until the that last followup.
The above was revised and added to the case presentation.
Question 5:
Why was CT scan not performed?
Response 5:
The CT scan was performed preoperatively for the evaluation. In the discussion, we mentioned that the intervertebral disc height was measured to be approximately 10.2 – 11.8 mm in the sagittal plane through computed tomography (not present in the article), which was matched to the size of the retropulsed cage used in the patient. However, the CT scan images were not presented in the manuscript since the radiographs and MRI scan provided enough and adequate information for the evaluation and preoperative planning. We hope not to make the article too lengthy. Thanks.
Reviewer 4 Report
The authors present an interesting case that merits publication after revision of several points:
- More information should be provided on the patient's preoperative course. The authors mention six previous surgeries. Which surgeries were performed, and for what pathologies?
- Although a rare complication of spinal fusion, cases of intradural cage migration have been described (e.g. doi/10.28982/josam.1010502), they should be discussed.
- Table 1 should be extensively revised and references added for the risk factors cited, incl. risk measurements, e.g. OR
- Supplementary Fig. is unnecessary as the urodynamic results are sufficiently discussed in the main text. In my opinion, a postoperative X-ray or ct would be more relevant to the readers.
Some sentences are a bit long, shortening those would likely improve readability.
Author Response
Thank you for your suggestions and appreciation of this report.
Question 1:
More information should be provided on the patient's preoperative course. The authors mention six previous surgeries. Which surgeries were performed, and for what pathologies?
Response 1:
Thank you for your suggestion. From the preoperative images, we speculated PLIF was done from L2 to L5 at least, as the radiographs showed the spinous process and lamina were resected. And, the removal of implant was done, as there was the trace of pedicle screw insertion at L5 and S1 bilaterally. The above was mentioned in the case presentation.
The patient reported low back pain and right-sided sciatica before the very first surgery in the local hospital. Unfortunately, we do not have the full documentation and images from the previous facility. Nonetheless, the first radiographs taken in our facility are still daunting and thought-provoking.
Question 2:
Although a rare complication of spinal fusion, cases of intradural cage migration have been described (e.g. doi/10.28982/josam.1010502), they should be discussed.
Response 2:
Thank you for your information. The case in the article (doi/10.28982/josam.1010502) revealed a retropulsed cage in L3/4 with partial encasement in the dura, which was similar but different to our patient. The symptoms of the patient and the degree of dural involvement differs. Though the material of the cage was not mentioned in that case, I suppose it was made of PEEK (from the picture of that article), which could be more indolent for its biocompatibility.
We revised the discussion as follows:
Intradural lesions from outside the dura mater are rare.6 One case report showed a case with retropulsed interbody cage at the L3/4 intervertebral area with partial intradural encasement.7 Though similar to our presented case, the symptoms and the degree of the dural involvement differed. The case presented here is remarkable due to the extreme rarity of complete intradural cage migration within the dura matter and the partial regain of the neurologic functions after more than two years.
Question 3:
Table 1 should be extensively revised and references added for the risk factors cited, incl. risk measurements, e.g. OR
Response 3:
Thank you for your suggestion. The table was modified. Please, re-check.
Question 4:
Supplementary Fig. is unnecessary as the urodynamic results are sufficiently discussed in the main text. In my opinion, a postoperative X-ray or ct would be more relevant to the readers.
Response 4:
Thank you for your helpful opinion. The urodynamic test is modified as figure 4. And we added the figure 5 for the immediate postoperative and one-year followup radiographs.
Figure.5
(A) and (B) The immediate postoperative anteroposterior and lateral views revealed extended pedicle screw fixation to L4 vertebrae, with removal of the intradural interbody cage. (C) and (D) After a year, the followup radiographs revealed no signs of implant failure or loosening.
Reviewer 5 Report
This report is a complication of spine surgery with implants; stray cage into the medulla is a very rare event. I consider this a didactic case report.
I am not sure I can answer some of the questions about the previous physician's treatment plan, but I did have numerous questions.
・Why are there multiple cages between L3/4 and 4/5 (conventionally there are often 1-2 cages)?
・Why were the pedicle screws and rods of L4-S1 removed?
I thought it would be best to add this information after obtaining information from the previous physician to the extent possible.
Also, the x-ray in Figure 1 does not show the cage deviating into the spinal canal (although it protrudes), whereas the MRI clearly shows the cage in the spinal canal.
・Has the cage moved due to the change in position?
・Is the magnetic field of the MRI affecting the cage?
・Is it because of the interval between imaging?
What do you think?
As discussed, inappropriate cage size selection may be a contributing factor, but I also thought that the cage may have migrated due to stress concentration between the L3/4 vertebrae as a result of the removal of posterior fixation at L3-S1. (Often, even though the intervertebral bodies appear to be fused, they are in fact not fully cross-linked.)
I thought it would be worth mentioning that at the very least, inadvertent implant removal can lead to subsequent implant-related complications.
Author Response

(The authors gave the same response as above.)

Round 2
Reviewer 3 Report
i have gone throug the manuscript including the response to reviewers comment . I would recommend acceptance of the manuscript .
Author Response
Thank you for your time and precious opinions.
Reviewer 4 Report
The authors have revised the manuscript in response to reviewer comments, however, regarding my previous review, two points still should be addressed:
1. Table 1 was not amended at all
2. As evident from the author's comments to another reviewer, the preoperative medical history (pervious surgeries, etc) was only partially available to the authors. I believe this should be mentioned in the case presentation
The authors should perform another critical language check, e.g. "After a year, the activities of daily living remained still... "
Author Response
Question 1:
Table 1 was not amended at all.
Response 1:
Table 1 is revised. Please, check.
Question 2:
The preoperative medical history (pervious surgeries, etc.) was only partially available to the authors. I believe this should be mentioned in the case presentation
Response 2:
Thank you for your precious opinion. We revised the case presentation as following:
A 43-year-old male, with a history of 6 lumbar surgeries for his lower back pain and right-sided sciatica at a local hospital, presented at our spine center with a 2-year-and-4-month history of incomplete paraplegia and cauda equina syndrome. The previous medical history was only partially available. He reported severe numbness in both lower limbs, and, mostly used a wheelchair for ambulation during the day.
Question 3:
The authors should perform another critical language check, e.g. "After a year, the activities of daily living remained still... "
Response 3:
Thank you. We will ensure a thorough critical language check is conducted.
As to the sentence, it was revised as following:
After one year, the patient’s activities of daily living remained unchanged (MRC grade 3-, ASIA grade D), and there was no significant complaints of numbness or back pain. Radiographs indicated that there were no signs of implant failures or loosening (Figure 5C and 5D).